# Gender bias in clinical trials of biological agents for migraine: A systematic review

**Marta Alonso-Moreno**[ID][☯], **Lupe Rodríguez-de Francisco**[ID][☯]*, **Pablo Ciudad-Gutiérrez**[ID][☯]

Pharmacy Unit, Virgen del Rocío University Hospital, Seville, Spain

☯ These authors contributed equally to this work.
* luperdefrancisco@gmail.com

## Abstract

Migraine is considered one of the most disabling diseases. Currently, there are few studies on clinical migraine treatment based on sex-related differences, despite the important role of sex in migraine. Our aim was to evaluate gender bias in published clinical trials on monoclonal antibodies (erenumab, galcanezumab, fremanezumab and eptinezumab). We performed a systematic review of controlled clinical trials of erenumab, galcanezumab, fremanezumab and eptinezumab, searching the PubMed/MEDLINE database for articles published before December 2021. The search identified 760 articles, 25 of which met the inclusion criteria. Of all the patients included in these trials, 85.1% were women. Only one study had female lead authors. Two of the 25 studies included a sex-based analysis of the primary endpoint. None of the articles discussed the results separately for men and for women. The proportion of men recruited in trials is scarce and more studies are needed to guarantee the safety and tolerability of monoclonal antibodies used in male migraine. As observed in our study, despite the high number of women recruited, only 2 studies analysed the results separately by sex. Thus, a potential risk of gender bias was found in these clinical trials.

## Introduction

Migraine is a neurological disorder that affects roughly 14% of people worldwide, and it is 2 or 3 times more common in women [1–4]. The main reason for this distinction appears to be the influence of female sex hormones, mainly estrogens [2, 5]. On the other hand, psychosocial factors such as a diagnosis of depression or anxiety, mostly suffered by women (prevalence of 25.1% and 9.6% respectively), and environmental factors could have an important impact on the presence of sex differences disparities in migraine prevalence [3, 6]. For this reason, it would be interesting to assess whether the results of clinical trials (CTs) also represent men, who are less affected by migraine.

According to the Global Burden of Disease Analysis, migraine is considered one of the most disabling diseases, and it affects both personal and professional qualities of life [4]. With respect to the frequency, duration and severity of migraine attacks, some studies have reported a considerable number of sex differences. Women show a greater number of attacks per month (>15 headaches) and experience progression to the chronic form of illness twice as

**Data Availability Statement:** All relevant data are within the paper and its Supporting information files.

**Funding:** The funders had no role in study design, data collection and analysis, decision to publish, or preparation of the manuscript.

**Competing interests:** The authors have declared that no competing interests exist.

often as men [3]. Moreover, migraine attacks are more likely to be severe and to be accompanied by additional symptoms in women, such as nausea and vomiting [1]. In contrast, symptoms develop earlier in men, with the average age of symptom onset at approximately 30–40 years, followed by a gradual rise [3, 5]. Additionally, a cross-sectional study showed that men with androgen deficiency, either to higher estradiol or lower testosterone levels suffer more frequently of migraine symptoms than males without a primary headache disorder [7].

Recently, biological agents have demonstrated a beneficial role in prophylactic and acute treatment of migraine. The goal is to reverse the effects of calcitonin gene related peptide (CGRP), which is theoretically responsible for causing an specific pain on the trigeminal system during migraine attacks [1]. There are few current studies on clinical migraine treatment based on sex-related differences, despite the important role of sex in migraine [1, 5]. Thus, some researchers suggest that future studies should consider sex differences, with a view to achieving more safe and effective migraine therapies [3].

Gender bias is a well documented problem detected in the design of clinical trials. The International Council for Harmonization (ICH) advises reporting subgroups when reporting the primary analyses in order to minimize any imbalance between subgroups [8]. In 1993, the Food and Drug Administration (FDA) issued guidance for the study and evaluation of sex differences in trials. Moreover, such guidance recommends the incorporation of an equal sample size of both sexes in the design and analysis of trials [9]. In 1994, the National Institutes of Health (NIH) published another guideline for the inclusion of minorities as clinical research subjects [10]. Another guideline was published by the Basque government in 2013, the Clinical Practice Guidelines Free of Gender Bias, which provides questions to aid in the identification of gender bias in health research [11].

Nevertheless, some systematic reviews on the design of clinical trials have shown gender bias. For instance, a systematic review of clinical trials about monoclonal antibodies (mAbs) for the treatment of multiple sclerosis, showed that 15 of the 55 trials analysed main variables separated by sex [12]. In another systematic review about biological agents for severe asthma, only 1 of the 37 clinical trials discussed results separated by sex [13]. In addition, in clinical trials of vortioxetine, the sex-separated analysis of the secondary variable was only performed in 6 of the 23 works included [14]. In clinical trials of diagnosed spondyloarthritis, women constituted only 32.3% of the total number of patients, revealing the existence of gender bias [15]. Similar cases have been observed in a systematic review of clinical trials on Irritable Bowel Syndrome, which is more common in women, where men were unrepresented (only 34,1% of patients) [16].

As mAbs are studied, evidence is increasing that the safety and tolerability of this class of migraine therapies are high and the adverse events are low [17]. There is a plan to develop additional mAbs for neurological diseases at a lower cost and with a better safety profile compared with current treatment options [18]. Thus, the aim of this article is to evaluate gender bias in published clinical trials on CGRP mAbs (erenumab, galcanezumab, fremanezumab and eptinezumab) authorised for prophylactic treatment of migraine. All are approved in the US by the FDA and are authorised in Europe by the European Medicines Agency (EMA).

## Methods

Our systematic review was performed in accordance with the Preferred Reporting Items for Systematic Reviews and Meta-Analyses Equity 2012 Extension declaration [19].

### Eligibility criteria

We selected the studies that met the following inclusion criteria:

- The study drug was erenumab, galcanezumab, fremanezumab or eptinezumab

- The clinical trials had a control group and random assignment

- Patients were adults (>18 years)

- Patients were diagnosed with episodic or chronic migraine

- The aim of the clinical trials was the evaluation of the efficacy and safety of the study drug. Clinical trials that additionally assessed other variables such as pharmacokinetic/pharmaco-dynamics were not excluded

   We excluded the following:

- Clinical trials in phase I

- *Post hoc* analyses of one or several previously published clinical trials and extension clinical tri-als of previously published trials. These articles included the same patients who were evaluated in the original articles. *Post hoc* analyses were considered secondary sources of information and discussed separately if the main variable or other variables that were not included in the analysis of the first publication of the original trial were analysed from a gender perspective

- Systematic reviews and meta-analyses

- Pilot studies with a small sample of patients (n<50)

- Short reports and letters to the editor

## Information sources

An electronic literature search was performed using PUBMED and EMBASE in December 2021, with no publication date or language restrictions. Search terms included a mixture of MeSH terms and free text (keywords and synonyms) combined with Boolean operators. The search strategy is detailed in Table 1. The references of selected studies were hand-searched to identify any other relevant studies.

## Study selection

Two independent reviewers (MAM and LRF) screened the titles and abstracts of all eligible publications for possible inclusion. To ensure inter-rater reliability, 100% of the articles were assessed independently by both authors. The articles included were fully read before a final decision on inclusion. Any disagreement was settled by consensus with a third reviewer (PCG).

**Table 1. Complete search strategy for different databases.**

| Healthcare Database | Search Strategy |
|---|---|
| PUBMED | (galcanezumab) OR (LY2951742) AND (migraine) AND (randomizedcontrolledtrial[Filter]) |
| | (erenumab) OR (AMG334) AND (migraine) AND (randomizedcontrolledtrial[Filter]) |
| | (fremanezumab) OR (TEV-4812) AND (migraine) AND (randomizedcontrolledtrial[Filter]) |
| | (epitinezumab) OR (ALD403) AND (migraine) AND (randomizedcontrolledtrial[Filter]) |
| EMBASE | galcanezumab:ab,ti AND migraine:ab,ti AND [randomized controlled trial]/lim |
| | erenumab:ab,ti AND migraine:ab,ti AND [randomized controlled trial]/lim |
| | fremanezumab:ab,ti AND migraine:ab,ti AND [randomized controlled trial]/lim |
| | eptinezumab:ab,ti AND migraine:ab,ti AND [randomized controlled trial]/lim |

## Data collection and analysis

The reviewers independently extracted data, and PCG examined all extraction sheets to ensure their accuracy. We explicitly stated whether any data were missing from clinical trials. For each publication, the following variables were registered:

- Drug under study

- Year of publication

- Mean age of patients (years)

- Financing of the trial: pharmaceutical industry or independent (the clinical trials were considered to be promoted by pharmaceutical companies if one of the authors was employed by a pharmaceutical company or if direct funding was specified)

- Location: US, Europe, Japan, Japan and Korea or the rest of the world.

- Trial phase

- Comparator: placebo or active drug

- Objectives of the trial: efficacy and safety, and if a pharmacodynamic/pharmacokinetic (PD/PK) evaluation was performed

- Diagnosis: episodic or chronic migraine

For the analysis of sex differences and to characterise the gender bias of the trials, we followed The Spanish recommendations for the study and evaluation of gender differences [11] and the FDA guide [9]. The methodology was also based on the ICH [8]. The following variables were analysed:

- Sex of the first author

- The number of patients recruited

- The number of women included and the percentage of women among the patients recruited

- Whether there were sex-stratified results of the primary or secondary outcomes

- Whether the concept of "gender" arose in the studies

- Whether the discussion of the results was analysed by sex

- Whether pregnancy was cited as an exclusion criterion

- Whether the studies analysed the interaction between hormone replacement therapy and the study drug, included women using hormonal contraceptives, analysed the interaction between hormonal contraceptives and the study drug, analysed the influence of the drug on the pharmacokinetics of hormonal contraceptives, investigated the effects of the phase of the menstrual cycle on the response to the drug, and studied the influence of the phase of the menstrual cycle on the pharmacokinetics of the drug.

## Results

As a result of the literature search, 760 studies were identified from various databases. Once the duplicates were eliminated using Mendeley checking, 435 studies were selected by title and abstract and 325 studies were deemed ineligible because they did not meet the inclusion criteria according to the PICOS criteria, mainly due to being a conference abstract or *post hoc*

analysis. A total of 75 articles were assessed in full text for eligibility; however, only 25 studies [20–44] complied with the inclusion criteria and were therefore included in this systematic review. Fig 1 shows the flowchart reflecting the election process of trials included.

Table 2 shows the variables that focused on the characteristics of the clinical trials included. Erenumab (8) was the most commonly assessed drug, followed by fremanezumab (6), galcanezumab (6) and eptinezumab (5). In the majority of clinical trials, the age range of the patients was 18–65 years (12), followed by 18–70 years (5), 18–55 years (2), 18–75 years (2), 20–65 years (2) and 18–60 years (1). All the articles were funded by pharmaceutical companies. The location of the clinical trials was worldwide in 15 studies, 4 in Japan, 4 in the US, and 2 in Japan and Korea; no study had been performed in Europe. Sixteen trials were performed in phase III, and 9 were performed in phase II. The comparator was placebo in all the articles. The studies evaluated efficacy and safety (16); efficacy, safety and PK (8); and efficacy, safety and PK/PD (1). The main diagnosis in 15 clinical trials was episodic migraine, followed by chronic migraine in 6 clinical trials and any type of migraine (episodic or chronic) in the 4 remaining studies.

Table 3 shows the variables that focus on possible gender bias. Only one study had a woman as the first author. The total number of patients recruited in clinical trials was 15,294; of these, women constituted 13,017 (85.1%). However, the analysis by sex of the primary outcome or secondary outcomes was only performed in 2 of the studies [43, 44]. No study discussed results by sex or took into consideration factors such as hormone replacement therapy, hormonal contraceptives or the menstrual cycle on the response or the pharmacokinetics related to the drugs studied. Nevertheless, pregnancy was an exclusion criterion in 8 clinical trials.

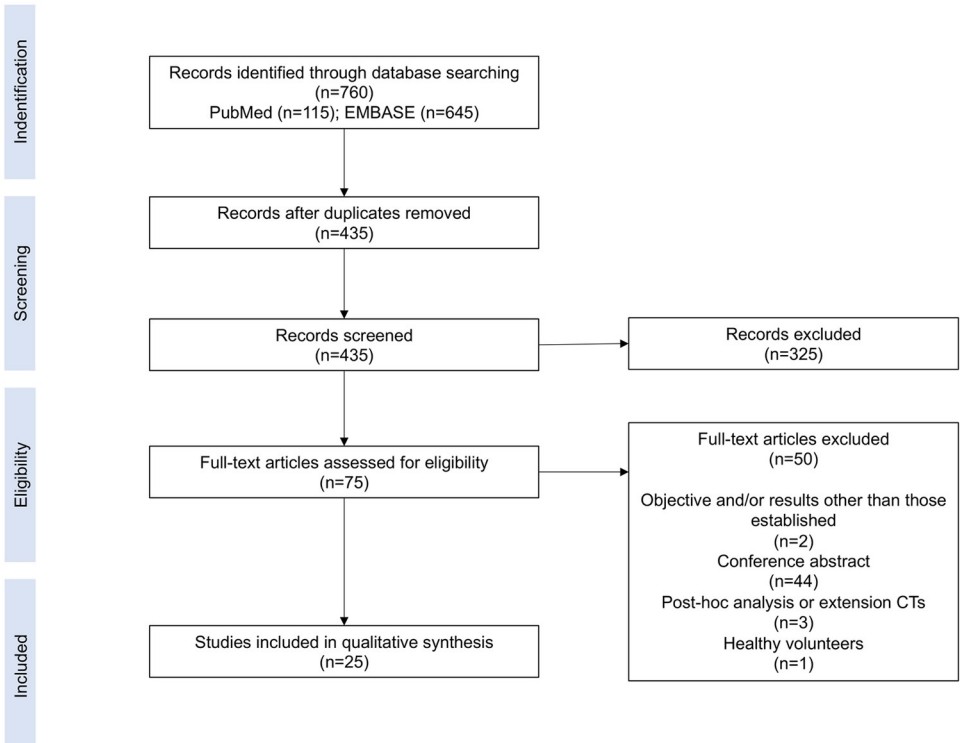

**Fig 1. Study selection flowchart.**

**Table 2. Characteristics of the included clinical trials.**

| Study | Drug | Mean age of patients (years) | Funding | CT location | Phase | Comparator | Objectives | Diagnosis |
|---|---|---|---|---|---|---|---|---|
| Goadsby et al, 2017 [20] | Erenumab | 18–65 | Ph. companies | ROW | III | Placebo | Efficacy and safety | Episodic migraine |
| Tepper et al, 2017 [21] | Erenumab | 18–65 | Ph. companies | ROW | II | Placebo | Efficacy, safety and PK | Chronic migraine |
| Dodick et al, 2018 [22] | Erenumab | 18–65 | Ph. companies | ROW | III | Placebo | Efficacy and safety | Episodic migraine |
| Reuter et al, 2018 [23] | Erenumab | 18–65 | Ph. companies | ROW | III | Placebo | Efficacy and safety | Episodic migraine |
| Sun et al, 2018 [24] | Erenumab | 18–60 | Ph. companies | ROW | II | Placebo | Efficacy, safety and PK | Episodic migraine |
| Sakai et al, 2019 [25] | Erenumab | 20–65 | Ph. companies | Japan | II | Placebo | Efficacy and safety | Episodic migraine |
| Goadsby et al, 2020 [26] | Erenumab | 18–65 | Ph. companies | ROW | III | Placebo | Efficacy and safety | Episodic migraine |
| Takeshima et al, 2021 [27] | Erenumab | 20–65 | Ph. companies | Japan | III | Placebo | Efficacy and safety | Episodic or chronic migraine |
| Dodick et al, 2014 [28] | Galcanezumab | 18–65 | Ph. companies | USA | II | Placebo | Efficacy, safety and PK | Chronic migraine |
| Oakes et al, 2018 [29] | Galcanezumab | 18–65 | Ph. companies | USA | II | Placebo | Efficacy, safety and PK/PD | Episodic migraine |
| Skljarevsky et al, 2018 [30] | Galcanezumab | 18–65 | Ph. companies | ROW | III | Placebo | Efficacy and safety | Episodic migraine |
| Mulleners et al, 2020 [31] | Galcanezumab | 18–75 | Ph. companies | ROW | III | Placebo | Efficacy and safety | Episodic or chronic migraine |
| Sakai et al, 2020 [32] | Galcanezumab | 18–65 | Ph. companies | Japan | II | Placebo | Efficacy and safety | Episodic migraine |
| Hirata et al, 2021 [33] | Galcanezumab | 18–65 | Ph. companies | Japan | III | Placebo | Efficacy, safety and PK | Episodic migraine |
| Dodick et al, 2014 [34] | Eptinezumab | 18–55 | Ph. companies | USA | II | Placebo | Efficacy, safety and PK | Episodic migraine |
| Dodick et al, 2019 [35] | Eptinezumab | 18–55 | Ph. companies | ROW | II | Placebo | Efficacy, safety and PK | Chronic migraine |
| Ashina et al, 2020 [36] | Eptinezumab | 18–75 | Ph. companies | ROW | III | Placebo | Efficacy, safety and PK | Episodic migraine |
| Lipton et al, 2020 [37] | Eptinezumab | 18–65 | Ph. companies | ROW | III | Placebo | Efficacy, safety and PK | Chronic migraine |
| Winner et al, 2021 [38] | Eptinezumab | 18–75 | Ph. companies | ROW | III | Placebo | Efficacy and safety | Episodic migraine |
| Bigal et al, 2016 [39] | Fremanezumab | 18–65 | Ph. companies | USA | II | Placebo | Efficacy and safety | Chronic migraine |
| Dodick et al, 2018 [40] | Fremanezumab | 18–70 | Ph. companies | ROW | III | Placebo | Efficacy and safety | Episodic migraine |
| Ferrari et al, 2019 [41] | Fremanezumab | 18–70 | Ph. companies | ROW | III | Placebo | Efficacy and safety | Episodic or chronic migraine |
| Goadsby et al, 2020 [42] | Fremanezumab | 18–70 | Ph. companies | ROW | III | Placebo | Efficacy and safety | Episodic or chronic migraine |
| Sakai et al, 2021 [43] | Fremanezumab | 18–70 | Ph. companies | Japan and Korea | III | Placebo | Efficacy and safety | Chronic migraine |
| Sakai et al, 2021 [44] | Fremanezumab | 18–70 | Ph. companies | Japan and Korea | III | Placebo | Efficacy and safety | Episodic migraine |

• Ph. companies = pharmaceutical companies, CT = clinical trial, ROW = rest of the world, USA = United States of America, PK = pharmacokinetic, PD = pharmacodynamic.

**Table 3. Proportion of women and other features of gender assessment.**

| Study | First author sex | Total patients | Total women | Percentage of women (%) | Analysis by sex of the main outcome | Analysis by sex of secondary outcomes | Discussed results analyzed by sex |
|-------|------------------|----------------|-------------|--------------------------|-------------------------------------|----------------------------------------|-------------------------------------|
| Goadsby et al, 2017 [20] | Male | 955 | 814 | 85.24 | No | No | No |
| Tepper et al, 2017 [21] | Male | 667 | 552 | 82.76 | No | No | No |
| Dodick et al, 2018 [22] | Male | 577 | 492 | 85.27 | No | No | No |
| Reuter et al, 2018 [23] | Male | 246 | 200 | 81.30 | No | No | No |
| Sun et al, 2018 [24] | Male | 483 | 389 | 80.54 | No | No | No |
| Sakai et al, 2019 [25] | Male | 475 | 400 | 84.21 | No | No | No |
| Goadsby et al, 2020 [26] | Male | 845 | 713 | 84.38 | No | No | No |
| Takeshima et al, 2021 [27] | Male | 261 | 227 | 86.97 | No | No | No |
| Dodick et al, 2014 [28] | Male | 217 | 184 | 84.79 | No | No | No |
| Oakes et al, 2018 [29] | Female | 410 | 340 | 83.00 | No | No | No |
| Skljarevsky et al, 2018 [30] | Male | 915 | 781 | 85.30 | No | No | No |
| Mulleners et al, 2020 [31] | Male | 462 | 397 | 85.93 | No | No | No |
| Sakai et al, 2020 [32] | Male | 459 | 387 | 84.31 | No | No | No |
| Hirata et al, 2021 [33] | Male | 311 | 268 | 86.17 | No | No | No |
| Dodick et al, 2014 [34] | Male | 163 | 133 | 81.59 | No | No | No |
| Dodick et al, 2019 [35] | Male | 616 | 535 | 86.99 | No | No | No |
| Ashina et al, 2020 [36] | Male | 888 | 749 | 84.30 | No | No | No |
| Lipton et al, 2020 [37] | Male | 1072 | 946 | 88.20 | No | No | No |
| Winner et al, 2021 [38] | Male | 480 | 403 | 83.96 | No | No | No |
| Bigal et al, 2016 [39] | Male | 261 | 227 | 86.97 | No | No | No |
| Dodick et al, 2018 [40] | Male | 875 | 742 | 84.80 | No | No | No |
| Ferrari et al, 2019 [41] | Male | 838 | 700 | 83.53 | No | No | No |
| Goadsby et al, 2020 [42] | Male | 1890 | 1645 | 87.04 | No | No | No |
| Sakai et al, 2021 [43] | Male | 571 | 491 | 85.99 | Yes | Yes | No |
| Sakai et al, 2021 [44] | Male | 357 | 302 | 84.59 | Yes | Yes | No |
| **Total** | **Women/Total 1/25** | **15294** | **13017** | **Mean = 85.11** | **2/25** | **2/25** | **0/25** |

**Table 4. Proportion of women and other features of gender assessment according to the different subgroups.**

| Subgroup | Studies | Representation of women | | | Analysis by sex | | |
|---|---|---|---|---|---|---|---|
| | N | N patients | N women | Percentage | Analysis by sex of the main outcome | Analysis by sex of secondary outcomes | Discussion of results by sex |
| | | | | | N/N Total Studies | N/N Total Studies | N/N Total Studies |
| Total | 25 | 15294 | 13017 | 85.1% | 2/25 | 2/25 | 0/25 |
| Geography | | | | | | | |
| USA | 4 | 1051 | 884 | 84.1% | 0/25 | 0/25 | 0/25 |
| Europe | 0 | 0 | 0 | 0% | 0/25 | 0/25 | 0/25 |
| Japan | 4 | 1506 | 1282 | 85.1% | 0/25 | 0/25 | 0/25 |
| Japan and Korea | 2 | 928 | 793 | 85.4% | 2/25 | 2/25 | 0/25 |
| ROW | 15 | 11809 | 10058 | 85.2% | 0/25 | 0/25 | 0/25 |
| Date of publication | | | | | | | |
| 2014–2017 | 5 | 2263 | 1910 | 84.4% | 0/25 | 0/25 | 0/25 |
| 2018–2021 | 20 | 13031 | 11107 | 85.2% | 2/25 | 2/25 | 0/25 |
| Outcome | | | | | | | |
| Efficacy+Safety | 16 | 10467 | 8921 | 85.2% | 2/25 | 2/25 | 0/25 |
| Efficacy+Safety+PK | 8 | 4417 | 3756 | 85.0% | 0/25 | 0/25 | 0/25 |
| Efficacy+Safety+PK/PD | 1 | 410 | 340 | 83.0% | 0/25 | 0/25 | 0/25 |
| Sample size | | | | | | | |
| N 0–200 | 1 | 163 | 133 | 81.6% | 0/25 | 0/25 | 0/25 |
| N 201–400 | 6 | 1653 | 1408 | 85.2% | 1/25 | 1/25 | 0/25 |
| N 401–700 | 10 | 5200 | 4386 | 84.3% | 1/25 | 1/25 | 0/25 |
| N 701–1000 | 6 | 5316 | 4499 | 84.6% | 0/25 | 0/25 | 0/25 |
| N +1000 | 2 | 2962 | 2591 | 87.5% | 0/25 | 0/25 | 0/25 |

ROW = rest of the world; PK = pharmacokinetic; PD: pharmacodynamic

Table 4 shows the proportion of women and other features of sex assessment according to the different subgroups. Twenty of the total trials were published between 2018 and 2021, whereas only 5 articles were published in previous years (2014–2017). One trial had fewer than 200 patients, with women representing 81.6%; 6 trials had 201–400 (85.2%), 10 had 401–700 (84.3%), 6 had 701–1000 (84.6%) and 2 had more than 1000 patients (87.5%).

## Discussion

On the one hand, our results show that, in all the clinical trials of erenumab, galcanezumab, fremanezumab and eptinezumab, the percentage of women was greater than 80%. These results are contrary to what mostly occurs in similar reviews, in which a low proportion of women have been recruited in trials [15, 16].

In this case, men are underrepresented in CTs probably because migraine is a disease much more prevalent in women [1–4]. However, recently it is tested that men with migraine and an androgen deficiency can suffer from severe symptoms, especially in those with chronic migraine [7]. Another study demonstrated that men are less likely to receive a diagnosis than women with migraine due to pronostic factors are well known in females [45]. For these reasons, it could be interesting to target equal sample sizes of men and women despite the fact that migraine affects more women than men.

On the other hand, there is an underrepresentation of women in scientific aspects. Only one included study had a woman as the first author of the publication [29]. The under-

representation of women as first authors is an issue well-described in the scientific literature, particularly in the neurology field. Giovannoni et al. highlighted the presence of gender bias on publication practices due to the low recognition of women in awards, editorial boards or primary scientific authorships, despite the greater number of successful female neurologists and neuroscientists worldwide [46]. According to a recent study, women remained in the minority of last (24.6%), first (36.2%) and middle (35.8%) author positions in articles published in 155 international neurology journals [47]. Another author suggested that the under-citation of women in neuroscience papers could be due to discriminatory values, practices and mechanisms that function in the domain of social institutions [48].

In terms of results, only 2 studies [43, 44] performed the analysis by sex of the main outcome or secondary outcomes. Although the limited funding of trials with a small sample size could explain the lack of analysis by sex, our analysis by subgroup found that no pharmaceutical industry-funded trials with a considerable sample size performed such analyses (see Table 4). In some fremanezumab *post hoc* studies, for the primary endpoint, analyses were performed that included sex as a covariate [49, 50]. However the results were not discussed by sex, which should be taken into account due to the low representation of men in the trials and to the variation of CGRP levels and hormones between sexes. Hormones affect CGRP, with a seemingly greater role for CGRP in females [1]. These aspects could be associated with differences in the response and efficacy of drugs interfering with the CGRP pathway.

Ornello et al. reported that 70% of women with migraine showed a menstrual association with their attacks and changes in headaches related to hormonal contraception, pregnancy and menopause in most headache centres. Thus, patients with menstrual-related migraine (MRM) account for about half of all migraineurs [2]. For this reason, there is an urgent need to educate healthcare professionals about sex-based differences in migraine and strategies for management of hormonally related headaches.

In our review, no clinical trial took into consideration factors such as hormone replacement therapy, hormonal contraceptives or the menstrual cycle on the response or the pharmacokinetics related to the drugs in the study, nor included the concept of gender in the text. This fact goes against the minimum requirements of scientific validity demanded by international guidelines and could be due to a possible attempt to minimise the total cost of trials [51, 52]. However, a *post hoc* study [53] of erenumab for prevention of episodic migraine (STRIVE) [26] determined the efficacy and safety of erenumab in women with self-reported menstrual migraine. Among 814 women enrolled in STRIVE, 232 (28.5%) reported a history of menstrual migraine and were aged ≤50 years. Of the 232 patients, 214 (92%) had a baseline monthly migraine of >5 days, suggesting a high proportion of women with attacks outside of the 5-day perimenstrual window (2 days before and 3 days after the start of menstruation). Results from this subgroup analysis of women with menstrual migraine were consistent with data from the overall STRIVE episodic migraine population, supporting the efficacy and safety of erenumab in women who experience menstrual migraine [53].

Pregnancy was mentioned only an exclusion criterion in 8 clinical trials. The rest of the trials did not mention it. During pregnancy, there is a theoretical risk for hypertension and foetal growth restriction when blocking CGRP. In pregnant rats, blocking CGRP with a CGRP receptor antagonist led to increased systolic blood pressure, foetal growth retardation and increased foetal mortality. It is curious that the increase in CGRP is measured during pregnancy; however, migraine patients typically experience fewer migraines during pregnancy, a difference that could be explained by variations in local cranial CGRP levels that increase during migraine and systemic CGRP, which plays a role in haemodynamic changes in pregnancy. Alternatively, high levels of CGRP in pregnancy could lead to a desensitisation of the receptor, leading to a decrease in migraines [1]. In the Goadsby et al. study, 6 women discontinued the

study due to a positive pregnancy test. Among these patients, 1 had a spontaneous abortion, 1 had a premature separation of the placenta (placental abruption) with no foetal loss, 1 had a premature baby and 1 had a foetal death. However, these events were assessed to be unrelated to fremanezumab by the investigator.

The main strength of this review is that it is the first systematic review that evaluates the concept of gender bias in the clinical trials of mAbs authorised in prophylactic treatment of migraine. We should also mention that the search was performed with no publication date or language restrictions in two of the largest healthcare databases, EMBASE and PUBMED, which incorporate articles published in renowned editorials and international journals.

## Conclusions

In conclusion, clinical trials have become a fundamental tool to evaluate the efficacy and safety of drugs; therefore, the study population should include a well-balanced proportion of women and men. In this case, the proportion of men recruited in trials is scarce (<20%), and more studies are needed to guarantee the safety and tolerability of mAbs used in men with migraine. Moreover, the inclusion of variables such as the stratification of results according to sex, and the influence of hormonal contraceptives on the pharmacokinetics of the studied drug should be fundamental requirements to publish an article in high impact journals. As observed in our study, despite the high number of women recruited in clinical trials, only 2 studies analysed the results separately by sex and men were underrepresented creating a large imbalance in CTs recruitment. Thus, a potential risk of gender bias was found in these clinical trials.

### Clinical implications

- The higher women's representation in anti-CGRPmAbs clinical trials reflect lack of knowledge on the efficacy and safety of these drugs in men.

- Stratification of results according to sex, and the influence of hormonal contraceptives on the pharmacokinetics of the studied drug should be fundamental requirements to publish an article in high impact journals.

- There is an urgent need to educate healthcare professionals about sex-based differences in migraine and strategies for management of hormonally related headaches.

## Supporting information

**S1 Checklist. PRISMA 2020 checklist.**
(PDF)

## Author Contributions

**Writing – original draft:** Marta Alonso-Moreno, Lupe Rodríguez-de Francisco, Pablo Ciudad-Gutiérrez.

**Writing – review & editing:** Marta Alonso-Moreno, Lupe Rodríguez-de Francisco, Pablo Ciudad-Gutiérrez.

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
