## [Decision Letter · Decision Letter 0]

1 Dec 2022

PONE-D-22-14914

GENDER BIAS IN CLINICAL TRIALS OF BIOLOGICAL AGENTS FOR MIGRAINE: A SYSTEMATIC REVIEW

PLOS ONE

Dear Dr. Rodríguez-de Francisco,

Thank you for submitting your manuscript to PLOS ONE. After careful consideration, we feel that it has merit but does not fully meet PLOS ONE’s publication criteria as it currently stands. Therefore, we invite you to submit a revised version of the manuscript that addresses the points raised during the review process.

A rebuttal letter that responds to each point raised by the academic editor and reviewer(s). You should upload this letter as a separate file labeled 'Response to Reviewers'.A marked-up copy of your manuscript that highlights changes made to the original version. You should upload this as a separate file labeled 'Revised Manuscript with Track Changes'.An unmarked version of your revised paper without tracked changes. You should upload this as a separate file labeled 'Manuscript'

We look forward to receiving your revised manuscript.

Kind regards,

Alejandro Piscoya

Academic Editor

PLOS ONE

https://journals.plos.org/plosone/s/fileid=ba62/PLOSOne_formatting_sample_title_authors_affiliations.pdf.

Reviewers' comments:

Reviewer's Responses to Questions

**Comments to the Author**

1. Is the manuscript technically sound, and do the data support the conclusions?

Reviewer #1: Yes

Reviewer #2: Yes

2. Has the statistical analysis been performed appropriately and rigorously? 

Reviewer #1: N/A

Reviewer #2: Yes

3. Have the authors made all data underlying the findings in their manuscript fully available?

Reviewer #1: No

Reviewer #2: Yes

4. Is the manuscript presented in an intelligible fashion and written in standard English?

Reviewer #1: Yes

Reviewer #2: Yes

5. Review Comments to the Author

Reviewer #1: This is a largely a competently executed and useful synthesis and evaluation of the literature. There are a few point that would strengthen the document.

1. The authors must distinguish more carefully in the entire document between “sex” which is an embodied, biological phenomena versus “gender” which the set of social meanings, including biases and discrimination, that are associated with sex (and which vary across place an time). It would be good to define the terms up front and then go through the paper and adjust carefully which term is being used where – the usage is currently interchangeable and imprecise. Factors such as hormones, menstruation, etc may be assumed be linked to sex. Failing to adjust for to such factors is an example of gender bias be researchers (i.e. failing to understand and account for sex-specific disease processes) Inclusion in trials and first author identity is an issue of gender bias. The failure to disaggregate the impact of drugs by sex (biological, since we are looking and biologic endpoint and pharmacodynamics) is a manifestation of gender bias (failing to recognize the importance of sex-specific data for people with different biology.

2. As regards the first authors of papers being evaluated: the relevant construct is GENDER of the first author, not sex. Please correct this.

3. Similarly: How was the gender of the first author established? Was this an assumption based on name, on-line photographs, professional acquaintanceships? All of these methods risk misclassification. If the individuals were asked to confirm their gender as of the time of publication on the manuscript, please so state. If you did not ask, please strongly consider doing so in a way that allows for the possibility that some individuals may be nonbinary, third gender, agender, etc. If you choose not to ask, the possibility of misclassification must be acknowledged as a limitation, as well as the limitation inherent in assuming that all the authors have a binary gender identity.

4. Given the very strong importance of hormones and hormonal cycles in the migraine disease process, I think the authors would do well to add an additional conclusion that failure to disaggregate these trial data by sex and supplemental hormone usage also compromises a proper understanding of the impact of these drug on *women* as well as men. Have men pooled in the analysis muddies the understanding of efficacy for women as well as leaving us with insufficient data for men. It’s a very important oversight across the board.

5. It is frequently reported in migraine patient forums that anti-CGRP agents impact differently on menstrual and non-menstrual migraines in patients who experience both, and that many find the prophylactic benefit to be limited to their non-menstrual migraines. This may be worth noting in reflecting on the overall biases and limitation of the literature.

6. It is worth noting in the limitations that neither the overall body of literature or this review have accounted for migraine patients who are intersex (sex category) or transgender, nonbinary, agender, etc (gender categories). The assumption that sex and gender and binary categories is common in medical literature and also inaccurate. It entirely omits the experiences of patients who take HRT as gender-affirming therapy (many of whom, regardless of their sex assigned at birth, report that such HRT impacts either positively or negatively the frequency and severity of their migraine disease).

Reviewer #2: The paper is good and it initiates other researchers and practitioners to consider the gender aspects of their study and and activities. However, it is good the researcher described the characteristics of included trails and proportion of women and features of gender assessment in concise manner.

6. PLOS authors have the option to publish the peer review history of their article (what does this mean?). If published, this will include your full peer review and any attached files.

Reviewer #1: No

Reviewer #2: **Yes: **Fentahun Gebrie Mucha

---

## [Author Response · Author response to Decision Letter 0]

26 Dec 2022

Reviewer #1: 

This is a largely a competently executed and useful synthesis and evaluation of the literature. There are a few point that would strengthen the document.

1. The authors must distinguish more carefully in the entire document between “sex” which is an embodied, biological phenomena versus “gender” which the set of social meanings, including biases and discrimination, that are associated with sex (and which vary across place an time). It would be good to define the terms up front and then go through the paper and adjust carefully which term is being used where – the usage is currently interchangeable and imprecise. Factors such as hormones, menstruation, etc may be assumed be linked to sex. Failing to adjust for to such factors is an example of gender bias be researchers (i.e. failing to understand and account for sex-specific disease processes) Inclusion in trials and first author identity is an issue of gender bias. The failure to disaggregate the impact of drugs by sex (biological, since we are looking and biologic endpoint and pharmacodynamics) is a manifestation of gender bias (failing to recognize the importance of sex-specific data for people with different biology.

Author response: We agree with the reviewer’s assessment. Accordingly, throughout the manuscript, we have revised more carefully the use of both terms and we have added both definitions, line 78:

“The term "sex" refers to a set of biological attributes that are associated with genetic expression, while the term "gender" refers to the socially constructed roles, attitudes and identities of individuals.”

2. As regards the first authors of papers being evaluated: the relevant construct is GENDER of the first author, not sex. Please correct this.

Author response: Thank you for pointing this out. The reviewer is correct, and we have replaced the word “sex” by “gender”. The revised text reads as follows on:

Line 181: “Gender of the first author”

Line 253: “First author gender”

3. Similarly: How was the gender of the first author established? Was this an assumption based on name, on-line photographs, professional acquaintanceships? All of these methods risk misclassification. If the individuals were asked to confirm their gender as of the time of publication on the manuscript, please so state. If you did not ask, please strongly consider doing so in a way that allows for the possibility that some individuals may be nonbinary, third gender, agender, etc. If you choose not to ask, the possibility of misclassification must be acknowledged as a limitation, as well as the limitation inherent in assuming that all the authors have a binary gender identity.

Author response: We agree that this is a potential limitation of the study. We have added this as a limitation on the discussion part, line 302:

“However, a potential limitation is the possibility of misclassification of the first author's gender because it was a name-based assumption as well as the limitation in assuming that all the authors have a binary gender identity."

4. Given the very strong importance of hormones and hormonal cycles in the migraine disease process, I think the authors would do well to add an additional conclusion that failure to disaggregate these trial data by sex and supplemental hormone usage also compromises a proper understanding of the impact of these drug on *women* as well as men. Have men pooled in the analysis muddies the understanding of efficacy for women as well as leaving us with insufficient data for men. It’s a very important oversight across the board.

Author response: We totally agree with you, and it is clear that failure to disaggregate these trial data by sex and supplemental hormone usage is one of the most interesting points to continue working on. We have tried to specify a little more final conclusions according to your comment, line 365:

“In conclusion, clinical trials have become a fundamental tool to evaluate the efficacy and safety of drugs; therefore, the study population should include a well-balanced proportion of women and men. In this case, the proportion of men recruited in trials is scarce (<20%), and more studies with sufficient data are needed to guarantee the safety and tolerability of mAbs used in men with migraine. Moreover, the inclusion of variables such as the stratification of results according to sex, and the influence of hormonal contraceptives on the pharmacokinetics of the studied drug should be fundamental requirements to publish an article in high impact journals because compromises a proper understanding of the impact of these drug on women as well as men. As observed in our study, despite the high number of women recruited in clinical trials, only 2 studies analysed the results separately by sex and men were underrepresented creating a large imbalance in CTs recruitment in addition to muddy the understanding of efficacy for women. Thus, a potential risk of gender bias was found in these clinical trials.”

5. It is frequently reported in migraine patient forums that anti-CGRP agents impact differently on menstrual and non-menstrual migraines in patients who experience both, and that many find the prophylactic benefit to be limited to their non-menstrual migraines. This may be worth noting in reflecting on the overall biases and limitation of the literature.

Author response: As you pointed out, the prophylactic benefit of anti-CGRP agents only on non-menstrual migraines is a reflection that we have added as a limitation of the literature, line 323:

“Moreover, this issue could be a limitation of the literature due to the potential benefits of anti-CGRP agents on non-menstrual migraines in women who experience both, menstrual and non-menstrual migraines”.

6. It is worth noting in the limitations that neither the overall body of literature or this review have accounted for migraine patients who are intersex (sex category) or transgender, nonbinary, agender, etc (gender categories). The assumption that sex and gender and binary categories is common in medical literature and also inaccurate. It entirely omits the experiences of patients who take HRT as gender-affirming therapy (many of whom, regardless of their sex assigned at birth, report that such HRT impacts either positively or negatively the frequency and severity of their migraine disease).

Author response: We totally agree with you that it is relevant to highlight the experiences of migraine intersex or transgender patients who are taking HRT, line 339:

“However, a potential publication bias of this review is that migraine intersex or transgender patients who are taking hormone replacement therapy should be into account to assess gender-affirming therapy”.

Reviewer #2: 

The paper is good and it initiates other researchers and practitioners to consider the gender aspects of their study and and activities. However, it is good the researcher described the characteristics of included trails and proportion of women and features of gender assessment in concise manner.

Author response: Thank you very much for your positive comments towards our work. Our main goal is to encourage other researchers to consider gender aspects in their studies. 

However, we believe that we have concisely described the characteristics of clinical trials. As we described in materials and methods we take into account for each publication, the following variables: drug under study, year of publication, mean age patients, financing of the trial, location, trial phase, comparator, objectives of the trial and diagnosis (Table 2).

In relation to the proportion of women and features of gender assessment, we have also taken into account variables that focus on the possible gender bias of each study as can be seen in Table 3 and which we have subsequently discussed and compared with other studies. 

If you have any suggestions for adding any more variables, we would be happy to add them to the manuscript to make it more complete.

---

## [Decision Letter · Decision Letter 1]

17 May 2023

GENDER BIAS IN CLINICAL TRIALS OF BIOLOGICAL AGENTS FOR MIGRAINE: A SYSTEMATIC REVIEW

PONE-D-22-14914R1

Dear Dr. Rodríguez-de Francisco,

We’re pleased to inform you that your manuscript has been judged scientifically suitable for publication and will be formally accepted for publication once it meets all outstanding technical requirements.

Kind regards,

Alejandro Piscoya

Academic Editor

PLOS ONE

Additional Editor Comments (optional):

Reviewers' comments:

Reviewer's Responses to Questions

**Comments to the Author**

1. If the authors have adequately addressed your comments raised in a previous round of review and you feel that this manuscript is now acceptable for publication, you may indicate that here to bypass the “Comments to the Author” section, enter your conflict of interest statement in the “Confidential to Editor” section, and submit your "Accept" recommendation.

Reviewer #2: All comments have been addressed

2. Is the manuscript technically sound, and do the data support the conclusions?

Reviewer #2: Yes

3. Has the statistical analysis been performed appropriately and rigorously? 

Reviewer #2: Yes

4. Have the authors made all data underlying the findings in their manuscript fully available?

Reviewer #2: Yes

5. Is the manuscript presented in an intelligible fashion and written in standard English?

Reviewer #2: Yes

6. Review Comments to the Author

Reviewer #2: It will be a base for other researchers in the area and let institutions use this findings to consider.

7. PLOS authors have the option to publish the peer review history of their article (what does this mean?). If published, this will include your full peer review and any attached files.

Reviewer #2: **Yes: **Fentahun Gebrie Mucha

---

## [Editor Report · Acceptance letter]

22 May 2023

PONE-D-22-14914R1 

Gender bias in clinical trials of biological agents for migraine: A systematic review 

Dear Dr. Rodríguez-de Francisco:

I'm pleased to inform you that your manuscript has been deemed suitable for publication in PLOS ONE. Congratulations! Your manuscript is now with our production department. 

Kind regards, 

on behalf of

Professor Alejandro Piscoya 

Academic Editor

PLOS ONE